# Generation and Characterization of a CE1-Modified mCherry-Expressing Influenza A Virus for In Vivo Imaging and Antiviral Drug Evaluation

**DOI:** 10.3390/v17121537

**Published:** 2025-11-24

**Authors:** Zhenghao Li, Meiyi Liu, Jia Yang, Qihui Sun, Dongxue Ye, Wanhui Zhou, Ruikun Du, Shijuan Cheng, Rong Rong, Yong Yang, Xiaoyun Liu

**Affiliations:** 1Innovative Institute of Chinese Medicine and Pharmacy, Shandong University of Traditional Chinese Medicine, Jinan 250355, China; 15615645631@163.com (Z.L.); liumeiyi698@163.com (M.L.); 2Experimental Center, Shandong University of Traditional Chinese Medicine, Jinan 250355, China; yangjia0721@163.com (J.Y.); yedongxuesss@163.com (D.Y.); 3Key Laboratory of Traditional Chinese Medicine Classical Theory, Ministry of Education, Shandong University of Traditional Chinese Medicine, Jinan 250355, China; sunqihui1214@163.com; 4College of Pharmacy, Shandong University of Traditional Chinese Medicine, Jinan 250355, China; 5Traditional Chinese Medicine Research Institute, Shandong Wohua Pharmaceutical Technology Co., Ltd., Weifang 261205, China; zhou_wanhui@163.com (W.Z.); chengs123@126.com (S.C.); 6Qingdao Academy of Chinese Medical Sciences, Shandong University of Traditional Chinese Medicine, Qingdao 266122, China; ruikun@sdutcm.edu.cn

**Keywords:** influenza A virus, CE1 mutation, mCherry expression, in vivo imaging, antiviral efficacy assessment

## Abstract

Influenza reporter viruses are essential for studying viral infection dynamics and assessing antiviral drug efficacy. However, insertion of exogenous reporter genes can impair both viral replication and reporter expression, limiting the development of these systems. In this study, CE1 compensatory mutation *(G3A*/*C8U)* was introduced into the 3′ non-coding region of the NS segment of influenza A/Puerto Rico/8/1934 using reverse genetics, generating the recombinant reporter virus H1N1-PR8-NS_CE1_-mCherry. Compared with H1N1-PR8-NS_WT_-mCherry, H1N1-PR8-NS_CE1_-mCherry produced approximately 2.7-fold more infectious particles. CE1 compensatory mutation partially restored impaired replication kinetics in vitro, as evidenced by higher titers of H1N1-PR8-NS_CE1_-mCherry at 48 h post-infection in MDCK cells. Additionally, H1N1-PR8-NS_CE1_-mCherry maintained the intact mCherry gene insertion and high viral titers during serial passaging. Additionally, a real-time, non-invasive in vivo imaging of influenza A viruses was established using H1N1-PR8-NS_CE1_-mCherry. A significant correlation was observed between lung fluorescence intensity and viral load, indicating that fluorescence signals serve as a reliable indicator of lung viral load in infected mice. Finally, utility of this model for in vivo drug screening was confirmed by antiviral drug oseltamivir phosphate. Above all, H1N1-PR8-NS_CE1_-mCherry provides a tool for visualizing influenza A virus infection and evaluating antiviral drug efficacy.

## 1. Introduction

Influenza A virus (IAV) is an enveloped virus of the *Orthomyxoviridae* family. Its genome comprises eight single-stranded, negative-sense RNA segments that encode hemagglutinin (HA), neuraminidase (NA), polymerase (PB2, PB1 and PA), nucleoprotein (NP), matrix protein (M) and nonstructural protein (NS) [1]. As a pathogen that causes seasonal epidemics and occasional pandemics in humans, IAV remains a major global public health threat [2,3,4,5]. While multiple FDA-approved antiviral drugs are available, including M2 ion channel blockers, neuraminidase inhibitors and cap-dependent endonuclease inhibitors, their extended use has facilitated the development of drug-resistant variants of IAV [6,7]. Furthermore, conventional antiviral screening typically relies on endpoint assays or destructive sampling, which precludes real-time, dynamic monitoring of therapeutic interventions. Consequently, accurate real-time monitoring systems for IAV may be essential for evaluating the effectiveness of antiviral therapies [8,9].

Recombinant IAVs expressing reporter genes provide the dynamic detection of viral infections and quantitative assessment of viral replication. For in vivo imaging, bioluminescent reporter gene imaging enables evaluations of intrinsic biological processes in live organisms, such as *Firefly* luciferase [10], *Renilla*/*Gaussia* luciferases [11] and *NanoLuc* [12]. Bioluminescent reporter systems generate light through enzyme-substrate reactions without requiring external excitation light and therefore typically offer high detection sensitivity with low background noise. Among these, *NanoLuc* shows particularly strong performance in deep-tissue imaging, combining high signal intensity with very low background, and its sensitivity is significantly higher than that of most conventional luciferases [13,14]. However, bioluminescent reporter systems typically emit monochromatic light, which limits their utility for multichannel or multiplex labeling. Furthermore, luciferase substrates must be administered via invasive routes such as injection, increasing animal discomfort, and their relatively high cost substantially raises overall experimental expenses. Compared with bioluminescent reporter systems, fluorescent protein-based imaging systems (e.g., GFP [15] and mCherry [16]) do not require substrate administration, simplify experimental workflows and enable truly non-invasive longitudinal imaging. Furthermore, fluorescent protein-based imaging systems can be readily combined with other imaging modalities to achieve multicolor visualization of diverse biological parameters. Among these, mCherry is widely used for live-animal imaging because its longer emission wavelength reduces tissue autofluorescence [17,18].

In influenza reporter viruses, mCherry gene is most commonly engineered into the NS segment, typically as an NS1 fusion protein or as an additional open reading frame immediately adjacent to nuclear export protein (NEP) [19]. By contrast, insertions of mCherry gene into the polymerase PA segment are less frequent [20]. However, published studies indicate that introducing fluorescent protein reporter genes into IAV impairs viral replication, compromises viral genetic stability, and leads to loss of reporter genes or mutations during serial passage or in vivo replication [21,22]. Consequently, improving the replication efficiency and enhancing the expression of reporter genes are critical for the development of in vivo imaging models of IAV and the assessment of antiviral drugs.

Non-coding regions (NCRs) play essential roles in the viral life cycle of IAV by regulating viral replication and transcription to maintain balanced gene expression and support efficient viral propagation [22,23,24,25]. Furthermore, sequence and structural variations in NCRs significantly influence viral adaptability and replication efficiency [26,27]. In recombinant IAVs expressing bioluminescent reporter genes, Zhao et al. introduced two mutations in the NS 3′-NCR, CE1 (*G3A*/*C8U*) and CE2 (*G3A*/*U5C*/*C8U)*, to enhance vRNA replication and transcription and thereby restore replication–transcription balance following reporter gene insertions. CE1 mutation provides a moderate compensatory enhancement suitable for small-to-medium inserts, whereas CE2 mutation offers a stronger compensatory effect that can rescue and stably propagate large inserts but may overcompensate for small inserts [10].

Therefore, mCherry gene was inserted into the NS segment of influenza A/Puerto Rico/8/1934 (H1N1-PR8), and CE1 mutation was introduced to generate H1N1-PR8-NS_CE1_-mCherry in this study. Using H1N1-PR8-NS_CE1_-mCherry, a real-time, non-invasive in vivo imaging platform was established to monitor IAV infection dynamics in mouse models. Furthermore, sensitivity and reliability of in vivo imaging platforms for antiviral drug screening were demonstrated by oseltamivir phosphate.

## 2. Materials and Methods

### 2.1. Cell Culture

Human embryonic kidney 293T cells and Madin-Darby canine kidney (MDCK) cells were cultured in Dulbecco’s modified Eagle medium (DMEM; Procell, Wuhan, China) containing 10% fetal bovine serum (Vivacell, Shanghai, China), 100 U/mL penicillin (Gibco, Carlsbad, CA, USA) and 100 μg/mL streptomycin (Gibco, Carlsbad, CA, USA). All cells were maintained at 37°C and 5% CO_2_. For viral infection, MDCK cells were incubated in influenza viruses isolated serum free medium (Yocon, Beijing, China) containing 1.5 μg/mL TPCK-treated trypsin (Sigma, St. Louis, MO, USA).

### 2.2. Plasmid Construction

Plasmids (pDZ-PR8-PB2, pDZ-PR8-PB1, pDZ-PR8-PA, pDZ-PR8-HA, pDZ-PR8-NA, pDZ-PR8-M, pDZ-PR8-NP and pPOLI-NS_CE1_-Gluc) containing genome of H1N1-PR8 were provided by Dr. Ruikun Du. Plasmid pcDNA3.1-PGK-mCherry was provided by Dr. Fujun Qin. All primer sequences used in this study were listed in Appendix A.

NS vector sequences containing CE1 mutation were amplified from pPOLI-NS_CE1_-Gluc using primers pPOLI-NS-F and pPOLI-NS-R. mCherry gene fragment was amplified from pcDNA3.1-PGK-mCherry using primers mCherry-F and mCherry-R. Plasmid pPOLI-PR8-NS_CE1_-mCherry was constructed by homologous recombination of NS vector sequences and mCherry gene fragments using ClonExpress Ultra One Step Cloning Kit V2 (Vazyme, Nanjing, China).

To generate plasmid pPOLI-PR8-NS_WT_-mCherry containing the wild-type NCR, nucleotides (*T3C* and *A8G*) of the NCR in pPOLI-PR8-NS_CE1_-mCherry were mutated using Mut Express II Fast Mutagenesis Kit V2 (Vazyme, Nanjing, China) with primers 3′ NCR-NS-F and 3′ NCR-NS-R.

Recombinant plasmids were transformed into DH5α competent cells (TSingKe, Beijing, China) according to the manufacturer’s protocol. Positive clones were screened by colony PCR and verified by DNA sequencing (TSingKe, Beijing, China).

### 2.3. Rescue and Amplification of Recombinant Reporter Viruses

293T cells (1 × 10^6^ cells/well) were seeded in 6-well plates and transfected upon reaching 50–60% confluency. Plasmid pPOLI-PR8-NS_WT_-mCherry or pPOLI-PR8-NS_CE1_-mCherry and seven other plasmids of H1N1-PR8 (pDZ-PR8-PB2, PB1, PA, HA, NA, M, NP; 500 ng) were co-transfected into 293T cells. MDCK cells (5 × 10^5^ cells/well) and TPCK-treated trypsin were added at 12 h post-transfection (hpt) and 36 hpt, respectively. Supernatants were collected at 48 hpt and inoculated 9-day-old specific pathogen-free (SPF) chicken embryos to amplify viruses.

### 2.4. Virus Growth Kinetics and In Vitro Fluorescence Detection

MDCK cells (5 × 10^5^ cells/well) in 6-well plates were infected with H1N1-PR8, H1N1-PR8-NS_WT_-mCherry or H1N1-PR8-NS_CE1_-mCherry at an MOI of 0.001 for 2 h at 37 °C. Then, supernatants were removed and influenza viruses isolated serum free medium containing TPCK-treated trypsin was added, followed by incubation at 37 °C. At the indicated times after infection, red fluorescent signals were monitored using a fluorescence microscope with a fixed 40 ms exposure time. Further, supernatants were collected for 50% tissue culture infectious dose (TCID_50_) assay.

### 2.5. TCID_50_ Assay

MDCK cells (5 × 10^3^ cells/well) in 96-well plates were infected with 10-fold serial dilution of viruses for 48 h at 37 °C, and then cytopathic effects (CPE) were recorded. TCID_50_ was calculated using the Reed–Muench method.

### 2.6. Genome Genetic Stability Analysis

To evaluate genetic stability of the inserted mCherry gene, a serial passage experiment of H1N1-PR8-NS_CE1_-mCherry was performed in chicken embryos. Briefly, 6.32 × 10^3^ TCID_50_ H1N1-PR8-NS_CE1_-mCherry was inoculated into 9-day-old SPF chicken embryo. Allantoic fluids were harvested at 48 h post-inoculation and used for TCID_50_ assay and RT-PCR.

### 2.7. RT-PCR

Viral RNA was extracted from allantoic fluids using TIANamp Viral RNA Kit (TianGen, Beijing, China) following the manufacturer’s protocol. mCherry gene fragment was amplified using primers (mCherry-F and mCherry-R) and HiScript^®^ II One Step RT-PCR Kit (Vazyme, Nanjing, China). PCR products were analyzed by agarose gel electrophoresis.

### 2.8. Determination of Median Lethal Dose (LD_50_)

To evaluate viral pathogenicity, 18–20 g male BALB/c mice were anesthetized with isoflurane, and intranasally inoculated with 20 μL 10-fold serial dilutions of either H1N1-PR8 or H1N1-PR8-NS_CE1_-mCherry. Body weight and survival were recorded daily for 14 days post-inoculation. Mice were euthanized when body weight loss exceeded 25%. LD_50_ was determined using nonlinear regression.

### 2.9. Establishment of a Robust Live Imaging Animal Model for IAV

To establish a live imaging model for IAV, male C57BL/6J mice (18–20 g) were randomly assigned by body weight to control or model groups. Under isoflurane anesthesia, mice in the control group received 20 μL saline intranasally, while mice in the model group were inoculated intranasally with 20 μL H1N1-PR8-NS_CE1_-mCherry (1 × 10^4^ TCID_50_). Mice were weighed daily for 14 days post-inoculation, and mCherry fluorescence signals were monitored at predefined time points using the Xenogen IVIS 200 imaging system (PerkinElmer, Waltham, USA) under isoflurane anesthesia. Data were analyzed using Living Image software (version 4.4).

### 2.10. Correlation Analysis Between Fluorescence Intensity and Viral Load

To investigate the correlation between fluorescence intensity and viral load, male C57BL/6J mice (18-20 g) were intranasally inoculated with 20 μL H1N1-PR8-NS_CE1_-mCherry (1 × 10^4^ TCID_50_) under isoflurane anesthesia. At predefined time points, mice were anesthetized with isoflurane and mCherry fluorescence signals were monitored using the Xenogen IVIS 200 imaging system. Then, mice were euthanized and lung tissues were excised for in vitro mCherry fluorescence imaging using the Xenogen IVIS 200 imaging system. Viral load in lung tissues was quantified by qPCR.

### 2.11. qPCR

Total RNA was extracted from lung tissues using the RNAprep Pure Tissue Kit (TianGen, Beijing, China), then reverse-transcribed to cDNA using the FastKing RT Kit with gDNase (TianGen, Beijing, China). qPCR was performed using SYBR Green reagents (TianGen, Beijing, China). Plasmid pDZ-PR8-M was served as the quantification standard. Viral load was quantified via a standard curve by converting CT values to viral genome copy numbers.

### 2.12. Oseltamivir Intervention

For antiviral assessment, male C57BL/6J mice (18-20 g) were randomly assigned by body weight to control, model and oseltamivir groups. Under isoflurane anesthesia, mice in the control group received 20 μL saline intranasally, whereas mice in the model and oseltamivir groups were inoculated intranasally with 20 μL H1N1-PR8-NS_CE1_-mCherry (1 × 10^4^ TCID_50_). Mice in the oseltamivir group were administered by oral gavage at 19.5 mg/kg oseltamivir phosphate. Mice in the control and model groups received distilled water. Treatment was administered once daily for 5 days, starting 1-day post-inoculation. mCherry fluorescence signals were monitored at predefined time points using the Xenogen IVIS 200 imaging system to assess antiviral efficacy.

### 2.13. Statistical Analysis

Data were expressed as mean ± SEM. Student’s *t*-test was used to determine statistical significance between two groups. Comparisons among three groups were performed by one-way ANOVA. Statistical significance was set at *p* < 0.05 (* *p* < 0.05, ** *p* < 0.01, *** *p* < 0.001, **** *p* < 0.0001).

## 3. Results

### 3.1. Generation of Recombinant IAV-Expressing mCherry

To generate recombinant IAV-expressing mCherry, mCherry coding sequences were inserted at nucleotide position 716 of NS segment (NS_WT_-mCherry; Figure 1A). In addition, PTV-1 2A peptide was used to facilitate co-translational separation of mCherry and NEP. Linker peptides (GSG) were inserted between NS1 and mCherry, as well as between mCherry and PTV-1 2A. To eliminate potential detrimental effects of mCherry insertion on viral replication, CE1 compensatory mutation (*G3A*/*C8U*) was introduced into the 3′-NCR of NS segment (NS_CE1_-mCherry; Figure 1A).

Subsequently, recombinant influenza reporter viruses H1N1-PR8-NS_WT_-mCherry and H1N1-PR8-NS_CE1_-mCherry were generated using the eight-plasmid reverse genetics system. Results showed that distinct red fluorescence signals were observed in MDCK cells at 24 hpt, which provided evidence for successful mCherry expression and virus rescue (Figure 1B). Then, supernatants were collected at 48 hpt and viral genome copy numbers were quantified by digital PCR. Results demonstrated that H1N1-PR8-NS_CE1_-mCherry exhibited significantly higher viral titers in the supernatants compared to H1N1-PR8-NS_WT_-mCherry. This finding suggested that CE1 mutation mitigated the detrimental effects of mCherry gene insertion on viral replication and enhanced viral production efficiency during the initial rescue phase (Figure 1C).

### 3.2. Characterization of H1N1-PR8-NS_WT_-mCherry and H1N1-PR8-NS_CE1_-mCherry

To systematically evaluate the effects of mCherry gene insertion and CE1 mutation on IAV biology, mCherry expression and replication efficiency of H1N1-PR8-NS_WT_-mCherry and H1N1-PR8-NS_CE1_-mCherry were compared. Results showed that H1N1-PR8-NS_CE1_-mCherry-infected cells exhibited significantly higher fluorescence intensity than H1N1-PR8-NS_WT_-mCherry-infected cells at all time points examined (Figure 2A), which indicated that CE1 mutation substantially enhanced mCherry expression. Although both recombinant viruses had the lower replication capacity than wild-type virus (H1N1-PR8), H1N1-PR8-NS_CE1_-mCherry exhibited higher replication efficiency than H1N1-PR8-NS_WT_-mCherry, reaching the titer of 10^8^ TCID_50_/mL at 60 h post-infection (hpi), suggesting that CE1 mutation partially restored viral replication capacity compromised by mCherry gene insertion (Figure 2B).

Given that H1N1-PR8-NS_CE1_-mCherry demonstrated significantly superior virus rescue efficiency, mCherry expression, and replication capacity in vitro compared to H1N1-PR8-NS_WT_-mCherry, it was selected as the optimized reporter virus and used for subsequent investigations.

### 3.3. Genetic Stability and Pathogenicity of H1N1-PR8-NS_CE1_-mCherry

To evaluate genetic stability of H1N1-PR8-NS_CE1_-mCherry, serial passages of H1N1-PR8-NS_CE1_-mCherry in chicken embryos were performed. Agarose gel electrophoresis confirmed the stable maintenance of mCherry gene without deletion in all passaged viruses (Figure 3A). Furthermore, the high viral titers were detected throughout all passaged viruses, indicating that CE1 mutation retained stable replicative capacity during serial propagation (Figure 3B). Taken together, these results demonstrated the stability of H1N1-PR8-NS_CE1_-mCherry over at least five consecutive passages.

Subsequently, pathogenicity of H1N1-PR8-NS_CE1_-mCherry was assessed. H1N1-PR8 demonstrated higher virulence, which caused significant weight loss and mortality in mice at low doses (LD_50_ = 11.87 TCID_50_). In contrast, H1N1-PR8-NS_CE1_-mCherry showed a LD_50_ of 5443 TCID_50_, indicating that mCherry gene insertion significantly attenuated viral virulence (Figure 3C). Nevertheless, H1N1-PR8-NS_CE1_-mCherry still caused notable weight loss and lethality of mice (Appendix A).

### 3.4. Establishment of a Robust Live Imaging Animal Model Using H1N1-PR8-NS_CE1_-mCherry

To dynamically monitor the spatiotemporal distribution and replication of IAV in vivo, an imaging model using H1N1-PR8-NS_CE1_-mCherry was established in mice. Initial attempts using BALB/c mice failed to detect fluorescent signals, likely due to limited tissue penetration of mCherry fluorescence and interference from the white fur background. Consequently, male C57BL/6J mice were used to established imaging model via intranasal inoculation with H1N1-PR8-NS_CE1_-mCherry (Figure 4A). Results indicated that mCherry fluorescent signals were first detected in the nasal region at 2 days post-infection (dpi), and subsequently detected in the lungs at 3 dpi. Fluorescence intensity in lung showed progressive increase at 3–6 dpi, reaching the plateau phase with sustained high levels during 7–9 dpi. The fluorescent intensity then began to decline, with signals diminishing to levels comparable to the control group at 12 dpi (Figure 4B and C). Concurrently, mice infected with H1N1-PR8-NS_CE1_-mCherry exhibited body weight loss patterns consistent with viral replication (Figure 4D).

Subsequently, correlation analysis between fluorescence intensity and viral load in mice lungs was quantified. Results demonstrated a significant association between fluorescence intensity and viral load in vivo (R^2^ = 0.48, *p* < 0.0001; Figure 5). In addition, correlation was stronger between fluorescence intensity and viral load in vitro (R^2^ = 0.69, *p* < 0.0001; Appendix A). These findings demonstrated that the H1N1-PR8-NS_CE1_-mCherry-based imaging model reflected viral replication dynamics in vivo and provided a method for real-time monitoring of IAV infection.

### 3.5. Application of H1N1-PR8-NS_CE1_-mCherry-Based Imaging Model for Antiviral Drug Screening

To validate H1N1-PR8-NS_CE1_-mCherry-based imaging model as a platform for antiviral drug evaluation, oseltamivir, which was a neuraminidase inhibitor that blocked viral particle release from infected cells, was selected as a representative compound (Figure 6A). Results indicated that model group exhibited significant weight loss, while the oseltamivir group showed attenuated weight loss (Figure 6B). These findings demonstrated that oseltamivir phosphate treatment significantly ameliorates clinical manifestations in mice infected with H1N1-PR8-NS_CE1_-mCherry. Additionally, oseltamivir phosphate treatment significantly reduced fluorescence intensity in lung at a clinically relevant dose (Figure 6C,D). These results indicated that the H1N1-PR8-NS_CE1_-mCherry-based imaging model provides an accurate method for assessing antiviral efficacy in vivo.

## 4. Discussion

Influenza reporter viruses are essential for probing infection dynamics and evaluating antiviral efficacy, because they enable real-time visualization of viral replication and dissemination. However, most recombinant reporter viruses exhibit attenuated replication capacity, which restrict their utility and reliability in vivo [21,22,28]. Therefore, improving the replication capacity of influenza reporter viruses and enhancing the expression of reporter genes remain major challenges in the field.

Recently, studies of influenza reporter viruses expressing fluorescent proteins have increased markedly. Common fluorescent proteins for in vivo imaging include GFP and mCherry. Among these, mCherry is most frequently used for live animal imaging, owing to its longer emission wavelength and lower autofluorescence background [17,18]. Nogales et al. used reverse genetics to generate a recombinant reporter virus ΔNS1-mCherry, in which mCherry gene replaced the PR8 NS1 coding sequences (28-690 AA) [21]. In addition, Bu et al. used A/Swine/Guangdong/GLW/2018 (H1N1) as the parental strain and generated GLW/18-MA-mCherry by inserting mCherry gene with PTV-1 2A at the PA C-terminus [20]. However, both ΔNS1-mCherry and GLW/18-MA-mCherry replicated less efficiently than wild-type virus. Consistent with these reports, H1N1-PR8-NS_WT_-mCherry also exhibited reduced replication efficiency (Figure 2B).

Several strategies have been employed to mitigate the reduction in viral replication associated with reporter genes insertion in recombinant influenza viruses. These include codon optimization of reporter genes, careful selection of genomic segment used for reporter genes insertion (e.g., PA, PB1 or NS) and balancing of promoter-driven replication and transcription through modifications in the NCRs [19,29,30]. To improve the replication capacity of influenza viruses expressing mCherry, a balance compensation strategy was adopted to construct H1N1-PR8-NS_CE1_-mCherry by introducing CE1 mutation into 3′-NCR of NS segment in this study. Compared with H1N1-PR8-NS_WT_-mCherry, CE1 mutation mitigated the mCherry gene insertion-associated replication defects and H1N1-PR8-NS_CE1_-mCherry showed a higher rescue rate (Figure 1C). Meanwhile, peak titers of H1N1-PR8-NS_CE1_-mCherry were approximately 10.79-fold higher than H1N1-PR8-NS_WT_-mCherry and about 3.78-fold lower than H1N1-PR8 in MDCK cells (Figure 2B). In parallel, CE1 mutation enhanced mCherry expression in vitro, producing stronger and earlier mCherry fluorescence signals. H1N1-PR8-NS_CE1_-mCherry produced detectable fluorescence signal at 12 hpi, whereas H1N1-PR8-NS_WT_-mCherry showed only a faint red signal even at 36 hpi (Figure 2A). Overall, H1N1-PR8-NS_CE1_-mCherry, which carries the CE1 mutation, shows greater replicative capacity and stronger mCherry expression than H1N1-PR8-NS_WT_-mCherry.

The reason for the greater replicative capacity and stronger mCherry expression of H1N1-PR8-NS_CE1_-mCherry may be that the CE1 mutation (*G3A/C8U*) altered terminal 3′-5′ base pairing, making viral RNA (vRNA)promoter more likely to form and stably maintain the pan-handle conformation. Pan-handle conformation enhances the recognition efficiency of RNA-dependent RNA polymerase, thereby promoting the initiation of vRNA replication and transcription and consequently compensating for the loss of viral replication capacity caused by the insertion of mCherry gene [10,31,32]. In addition to restoring the replication–transcription balance, we hypothesize that CE1-enhanced promoter activity in the modified NS segment may also impact innate immune regulation. Increased promoter activity could elevate NS1 expression, thereby suppressing type I interferon and interferon-stimulated gene (ISG) expression. At the same time, alterations in the stability of the 3′-NCR pan-handle structure may influence how vRNA was recognized by cytoplasmic pattern recognition receptors and affect the production of aberrant viral RNAs (such as defective interfering RNAs), which in turn modulate innate immune sensing [33,34,35]. Consistent with this, Zhao et al. showed that introducing CE1 or CE2 *(G3A*/*U5C*/*C8U)* mutations into 3′-NCR of NS gene in recombinant influenza viruses expressing Gluc could enhance viral RNA synthesis and help reporter viruses re-establish a replication–transcription balance that more closely resembles the physiological state [10]. CE2 mutation produces a stronger promoter-enhancing effect than CE1 mutation and is therefore a promising candidate to compensate larger exogenous gene insertions. Therefore, CE2 mutation was introduced into 3′-NCR to compare which CE2 or CE1 more effectively enhances mCherry expression in future. Given the conserved nature of 3′-NCR sequences, the balance compensation strategy may be applicable to other influenza virus subtypes (e.g., H3N2, H5N1 or H7N9). However, effectiveness of balance compensation strategy beyond the PR8 background requires experimental validation.

During model development, ΔNS1-mCherry failed to support fluorescence imaging in vivo, whereas GLW/18-MA-mCherry enabled real-time non-invasive fluorescence imaging. Based on H1N1-PR8-NS_CE1_-mCherry, we established an in vivo visualization model for IAV that exhibited a stable replication window, with peak fluorescence at 9 dpi. However, unlike GLW/18-MA-mCherry, H1N1-PR8-NS_CE1_-mCherry did not produce detectable fluorescence signals in BALB/c mice. Several factors may account for this discrepancy: (1) Differences in inoculum doses: H1N1-PR8-NS_CE1_-mCherry was administered at 1 × 10^4^ TCID_50_ (≈7 × 10^3^ PFU), whereas GLW/18-MA-mCherry was given at 1 × 10^5^ PFU. The lower inoculum of H1N1-PR8-NS_CE1_-mCherry may have produced weaker fluorescence signals, reducing detectability under the imaging conditions used. (2) Differences between imaging platforms: GLW/18-MA-mCherry and H1N1-PR8-NS_CE1_-mCherry were imaged using the PerkinElmer IVIS Lumina III and Xenogen IVIS 200, respectively. Variations in system sensitivity, filter sets and optical calibration between the two platforms may have affected signal detectability. (3) Sex-related differences in murine models: Female and male BALB/c mice were infected with GLW/18-MA-mCherry and H1N1-PR8-NS_CE1_-mCherry, respectively. Sex-dependent biological factors are likely to underlie the observed variability in imaging outcomes. Innate and adaptive immunity (for example, interferon signaling and recruitment of macrophages and neutrophils) were modulated by sex hormones, which can lead to changes in viral replication kinetics and thereby alter the timing and magnitude of mCherry expression.

Despite its utility, use of mCherry as a reporter gene for in vivo imaging has inherent limitations. Fluorescence signals of mCherry, while superior to GFP, have limited tissue penetration depth due to absorption and scattering by biological tissues [36]. This likely contributed to our initial failure to detect signals in BALB/c mice and limits the sensitivity for detecting deep-seated or low-level infections. Furthermore, sensitivity of mCherry was lower compared to near-infrared fluorescent proteins or luciferase-based bioluminescence systems [37]. To address these issues, future works should (1) use red-shifted or near-infrared reporters (e.g., iRFP) to improve tissue penetration, (2) optimize imaging hardware and acquisition settings to increase signal-to-noise ratio, (3) apply basic spectral unmixing or denoising and (4) enhance fluorescence signals by optimizing the expression of fluorescent proteins.

Using oseltamivir phosphate as a positive control, feasibility and reliability of the H1N1-PR8-NS_CE1_-mCherry-based mice model for visualizing antiviral drug efficacy were assessed in this study. The efficacy of other antiviral agents (e.g., polymerase cap-dependent endonuclease inhibitor baloxavir marboxil and M2 ion channel blocker amantadine) and traditional Chinese medicines that were used in this model should be explored in future studies. Moreover, combining this model with other fluorescent proteins or bioluminescent reporters could enable simultaneous detection of diverse biological signals, further expanding its applicability in antiviral drug research [16,38].

## 5. Conclusions

In summary, the mCherry-expressing influenza reporter virus H1N1-PR8-NS_CE1_-mCherry was constructed by introducing CE1 mutation (*G3A*/*C8U*) in this study. Compared with the uncompensated control H1N1-PR8-NS_WT_-mCherry, H1N1-PR8-NS_CE1_-mCherry exhibits enhanced viral replication efficiency and mCherry expression levels in vitro. Meanwhile, H1N1-PR8-NS_CE1_-mCherry was used to establish an in vivo visualization model of IAV for real-time viral infection monitoring and antiviral drug efficacy evaluation. These results suggested that CE1 mutation represents a promising strategy for generating influenza reporter viruses with enhanced viral replication efficiency and reporter gene expression, expanding its application prospects in monitoring influenza virus infection dynamics and assessing drug efficacy.

## Figures and Tables

**Figure 1 viruses-17-01537-f001:**
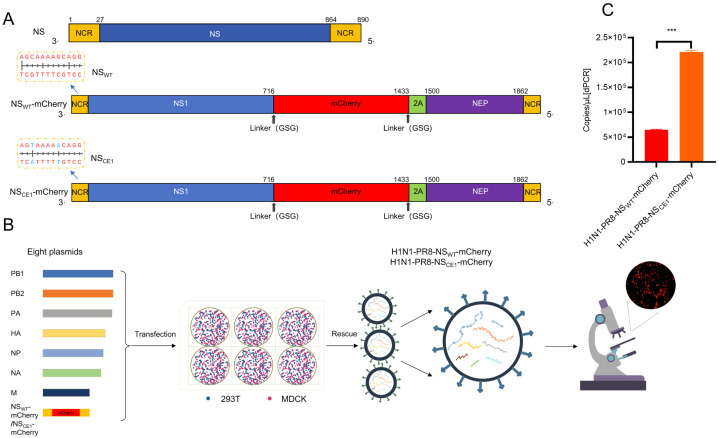
Generation of H1N1-PR8-NS_WT_-mCherry and H1N1-PR8-NS_CE1_-mCherry. (**A**) Schematic diagram of the natural and modified NS segment. NCR (yellow box), NS1 (blue box), NEP (purple box), mCherry (red box) and PTV-1 2A (green box) were shown. The introduced mutations at nucleotide positions 3 and 8 in the NCR were highlighted in blue. (**B**) Generation of reporter viruses by reverse genetics. (**C**) Quantification of viral gene copy numbers for reporter viruses by digital PCR. Data was presented as mean ± SEM of 2 independent replicates. *** *p* < 0.001, students’ *t* test.

**Figure 2 viruses-17-01537-f002:**
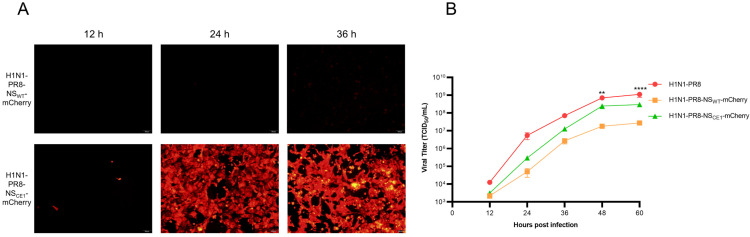
mCherry expression and growth kinetics of H1N1-PR8-NS_WT_-mCherry and H1N1-PR8-NS_CE1_-mCherry. (**A**) Red fluorescence signals of MDCK cells infected with H1N1-PR8-NS_WT_-mCherry or H1N1-PR8-NS_CE1_-mCherry. Scale bar: 100 μm. (**B**) Growth kinetics of H1N1-PR8, H1N1-PR8-NS_WT_-mCherry and H1N1-PR8-NS_CE1_-mCherry. Data was presented as mean ± SEM of 3 independent replicates. H1N1-PR8 vs. H1N1-PR8-NS_CE1_-mCherry: ** *p* < 0.01, **** *p* < 0.0001, one-way ANOVA.

**Figure 3 viruses-17-01537-f003:**
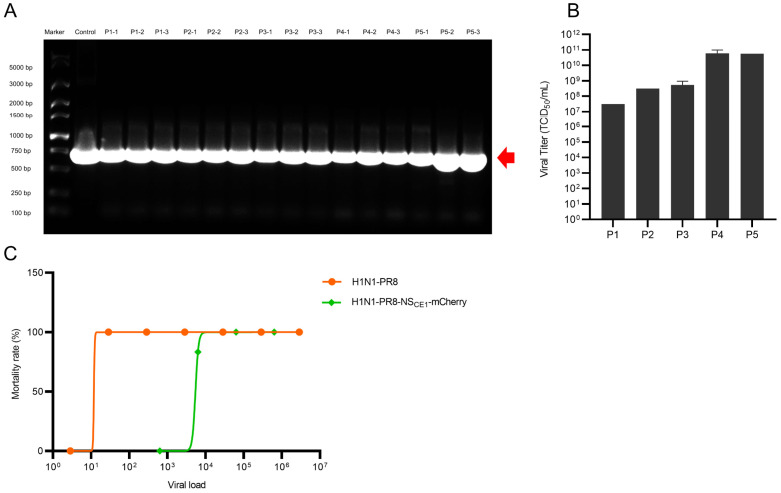
Genetic stability and pathogenicity of H1N1-PR8-NS_CE1_-mCherry. (**A**) PCR analysis of mCherry gene insertion in passaged H1N1-PR8-NS_CE1_-mCherry. The red arrow indicated the position of band representing mCherry gene. (**B**) Viral titer per generation. Data was presented as mean ± SEM of 3 independent replicates. (**C**) LD_50_ of H1N1-PR8 and H1N1-PR8-NS_CE1_-mCherry. *n* = 6.

**Figure 4 viruses-17-01537-f004:**
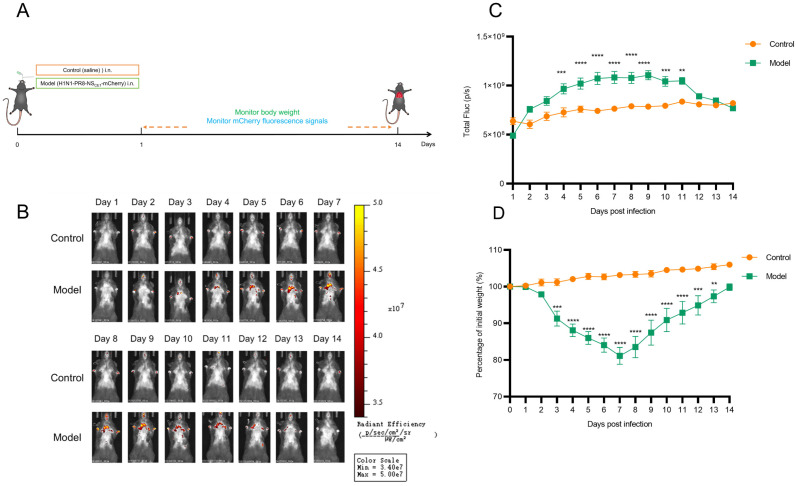
In vivo visualization of H1N1-PR8-NS_CE1_-mCherry in C57BL/6J mice. (**A**) Schematic illustration of the experimental design. (**B**) Representative in vivo fluorescence images of mice in each group. (**C**) Longitudinal changes in lung fluorescence intensity of mice in each group. *n* = 6, data was presented as mean ± SEM. ** *p* < 0.01, *** *p* < 0.001, **** *p* < 0.0001, students’ *t* test. (**D**) Body weight changes in mice in each group. *n*= 6, data was presented as mean ± SEM. ** *p* < 0.01, *** *p* < 0.001, **** *p* < 0.0001, students’ *t* test.

**Figure 5 viruses-17-01537-f005:**
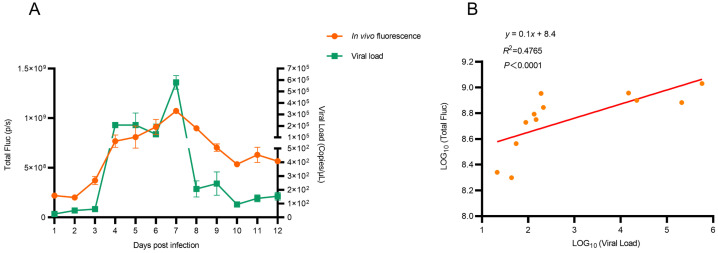
Correlation between in vivo fluorescence intensity and viral load. (**A**) In vivo fluorescence signals and viral load in lungs. *n*= 3, data was presented as mean ± SEM. (**B**) Correlation between in vivo fluorescence intensity and viral load.

**Figure 6 viruses-17-01537-f006:**
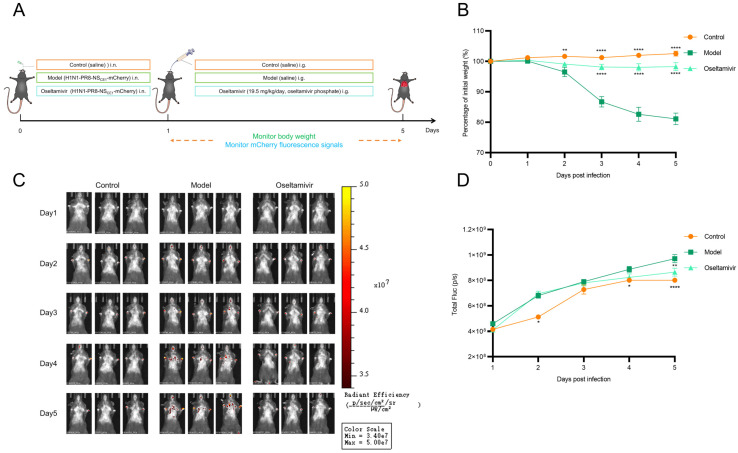
Application of the H1N1-PR8-NS_CE1_-mCherry in antiviral drug research. (**A**) Schematic illustration of the experimental design. (**B**) Body weight changes in mice in each group. *n* = 6, data was presented as mean ± SEM. Control vs. model: ** *p* < 0.01, **** *p* < 0.0001; oseltamivir vs. model: **** *p* < 0.0001, one-way ANOVA (**C**) Representative in vivo fluorescence images of mice in each group. (**D**) Quantification of fluorescence intensity of mice in each group. *n* = 6, data was presented as mean ± SEM. Control vs. model: * *p* < 0.05, **** *p* < 0.0001; oseltamivir vs. model: ** *p* < 0.01, one-way ANOVA.

## Data Availability

The data used to support the findings of this study are available from the corresponding authors upon reasonable request.

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
