# Peer review of "Generation and Characterization of a CE1-Modified mCherry-Expressing Influenza A Virus for In Vivo Imaging and Antiviral Drug Evaluation"

_viruses, 2025, doi:10.3390/v17121537_

Round 1
Reviewer 1 Report
Comments and Suggestions for Authors
The manuscript reports a stable recombinant influenza virus suitable for in vivo tracing: H1N1-PR8-NSCE1-mCherry. The study demonstrates that introducing the CE1 mutation enhances the genetic stability of the recombinant virus and yields a stronger fluorescent signal. This viral tracing system represents a potentially valuable tool for visualizing viral dynamics, which could aid in the development of antiviral drugs and vaccines.
Several points need to be addressed:
Major Concerns:
- The current study evaluates the stability of H1N1-PR8-NSCE1-mCherry only in embryonated chicken eggs. It would be important to also assess the stability of the recombinant virus during amplification in MDCK cells.
- It is recommended to include a control virus without the CE1 mutation (H1N1-PR8-NS-mCherry) in the stability assays to clearly demonstrate the effect of the CE1 mutation.
- The use of next-generation sequencing (NGS) is suggested to evaluate viral stability at the single-nucleotide level following amplification in eggs or cells, which would provide more detailed and rigorous data.
Minor Concerns:
- The font size in Fig. 1 is too small and should be enlarged for better readability.
- In Fig. 2A, the inclusion of a bright-field image or DAPI staining is recommended to confirm the number of cells per well.
- Fig. 4C and Fig. 6A currently lack statistical analyses; inclusion of appropriate significance tests is recommended
Author Response
|
Major Concerns: |
| Comments 1: The current study evaluates the stability of H1N1-PR8-NSCE1-mCherry only in embryonated chicken eggs. It would be important to also assess the stability of the recombinant virus during amplification in MDCK cells. |
| Response 1: Thank you for pointing this out. We agree with this comment. The stability of H1N1-PR8-NSCE1-mCherry in MDCK cells is currently being evaluated. However, the results are not yet available and therefore could not be incorporated into this revision. Additionally, we reviewed the literatures on stability of influenza virus [1,2], which indicated that chicken embryos serve as a commonly platform used for amplifying influenza viruses in both research and vaccine production. Compared with MDCK cells, chicken embryos typically produce higher viral titers and better preserve the genomic and antigenic features of clinical influenza virus isolates, thereby reducing the risk of culture-adapted mutations during in vitro propagation. Given our objective to preserve the native characteristics of H1N1-PR8-NSCE1-mCherry and minimize culture-induced mutations, chicken embryos were selected as the experimental systems in the initial study design. |
|
Comments 2: It is recommended to include a control virus without the CE1 mutation (H1N1-PR8-NS-mCherry) in the stability assays to clearly demonstrate the effect of the CE1 mutation. |
|
Response 2: Thank you for pointing this out. We agree with this comment. Addition of H1N1‑PR8‑NSWT‑mCherry in the stability assays would clearly demonstrate the effect of CE1 mutation on stability of influenza virus expressing mCherry. We have already initiated this experiment, but the results are not yet available and therefore could not be incorporated into this revision. Importantly, the primary objective of this study was to construct a recombinant influenza virus expressing mCherry and to establish an in vivo imaging model of influenza virus for antiviral drug evaluation, rather than to comprehensively delineate CE1 mutation on stability of influenza virus expressing mCherry. Therefore, we have revised the title to emphasize construction and application of H1N1‑PR8‑NSCE1‑mCherry. The revision can be found at the Title section - page 1, line 2-4 in the revised manuscript. |
|
Comments 3: The use of next-generation sequencing (NGS) is suggested to evaluate viral stability at the single-nucleotide level following amplification in eggs or cells, which would provide more detailed and rigorous data. |
|
Response 3: Thank you for pointing this out. We agree with this comment. We have now initiated this experiment. However, the results are not yet available and therefore could not be incorporated into this revision. Furthermore, the primary objective of this work was to construct a recombinant influenza virus carrying mCherry and to establish an in vivo imaging model of influenza virus for antiviral drug evaluation, rather than to comprehensively delineate CE1 mutation on stability of influenza virus expressing mCherry. Accordingly, the stability of mCherry gene was assessed by PCR and agarose gel electrophoresis in this study, and the results showed that mCherry gene was stably maintained over five consecutive passages. Based on these findings, we speculated that mCherry gene exhibits stable genetic characteristics in H1N1-PR8-NSCE1-mCherry. |
|
Minor Concerns: |
|
Comments 1: The font size in Fig. 1 is too small and should be enlarged for better readability. |
|
Response 1: Thank you for pointing this out. We agree with this comment. Therefore, we have revised Figure 1. The revision can be found at the following location - page 6, line 204, Figure 1 in the revised manuscript. |
|
Comments 2: In Fig. 2A, the inclusion of a bright-field image or DAPI staining is recommended to confirm the number of cells per well. |
|
Response 2: Thank you for pointing this out. We agree with this comment. Throughout the experiment, we strictly adhered to standard operating procedures (see Sections 2.1 and 2.4 in the Materials and Methods section) to ensure uniform cell seeding across all wells. Prior to fluorescence imaging, we assessed cell numbers and infection status using the bright‑field mode of the fluorescence microscope, in order to confirm that cell counts within the field of view were similar. However, we did not capture and archive these bright-field images to document cell morphology and confluency, which constitutes a limitation of this study. We will preserve bright-field images in future study. In this study, we adopted a longitudinal live‑cell imaging strategy, repeatedly observing the same cells at different time points rather than sacrificing separate cell samples at each time point. Therefore, DAPI and similar cell nuclei dyes, which predominantly stain the nuclei of fixed, permeabilized and thus non-viable cells, were not suitable for our experimental design. We also did not employ other membrane‑permeable cell nuclei dyes for live‑cell staining to prevent potential effects on cell viability or phenotype during the observation period. |
|
Comments 3: Fig. 4C and Fig. 6A currently lack statistical analyses; inclusion of appropriate significance tests is recommended. |
|
Response 3: Thank you for pointing this out. We agree with this comment. Therefore, we performed statistical significance tests on data of Figure 4C and 6A and added significance markers in the figures. The revision can be found at the following location - page 9, line 267 and 272-273, Figure 4C and page 10, line 297 and 299-301, Figure 6A in the revised manuscript. |
|
Reference 1. Hussain, S.; Turnbull, M. L.; Wise, H. M.; Jagger, B. W.; Beard, P. M.; Kovacikova, K.; Taubenberger, J. K.; Vervelde, L.; Engelhardt, O. G.; Digard, P., et al. Mutation of Influenza A Virus PA-X Decreases Pathogenicity in Chicken Embryos and Can Increase the Yield of Reassortant Candidate Vaccine Viruses. J Virol. 2019, 93. 2. Eisfeld, A. J.; Neumann, G.; Kawaoka, Y. Influenza A virus isolation, culture and identification. Nature Protocols. 2014, 9, 2663-2681.
|

Reviewer 2 Report
Comments and Suggestions for Authors
This study by Li et al. reports the development of a genetically stabilized influenza A reporter virus, H1N1-PR8-NSCE1-mCherry, achieved by introducing a CE1 compensatory mutation in the NS segment. The virus demonstrates improved replication, enhanced genetic stability, reliable in vivo fluorescence imaging, and suitability for antiviral drug evaluation. However, the manuscript requires substantial revision before publication.
Major points:
-
The authors should justify the choice of mCherry as the fluorescent reporter instead of luciferase or Nanoluciferase (Nluc). Several similar studies preferentially use Nluc viruses to minimize background signal in whole-animal imaging and to allow quantitative measurements (e.g., DOI: 10.1016/j.isci.2025.113402, DOI: 10.1128/spectrum.02150-25).
-
In Figure 3, while the authors demonstrate gene integrity over five passages using PCR and viral titers, they do not report whether mCherry expression is maintained or affected by viral adaptive mutations.
-
In Figures 4 and 6, fluorescent proteins are known to be influenced by the host background, as partially indicated in Figure 4. The authors should consider examining mCherry expression in ex vivo lung tissue.
-
Figure 4 lacks a comparison between H1N1-PR8-NSCE1-mCherry and H1N1-PR8-NSwt-mCherry, which would provide critical context for evaluating the impact of the CE1 mutation.
Minor points:
-
The resolution of several figures is low, particularly the nucleotide sequences in Figure 1, making detailed interpretation difficult.
Author Response
|
Major Concerns: |
|
Comments 1: The authors should justify the choice of mCherry as the fluorescent reporter instead of luciferase or Nanoluciferase (Nluc). Several similar studies preferentially use Nluc viruses to minimize background signal in whole-animal imaging and to allow quantitative measurements (e.g., DOI: 10.1016/j.isci.2025.113402, DOI: 10.1128/spectrum.02150-25). |
|
Response 1: Thank you for pointing this out. We agree with this comment. In this study, we selected mCherry as the fluorescent reporter gene for establishing an in vivo imaging model of influenza virus based on the following considerations. First, bioluminescence imaging requires prior injection of exogenous substrates (such as furimazine), whereas fluorescence imaging does not. Fluorescence imaging reduces procedural interference and animal distress, thereby enabling non‑invasive longitudinal in vivo imaging. Second, bioluminescent substrates are expensive, which markedly increases costs in studies involving a large number of animals. Third, bioluminescent systems generally provide only a single emission wavelength, limiting their suitability for multichannel or multiplex labeling. Forth, we plan to apply this model to multicolor imaging in subsequent study. The rationale and supporting literature have been added to the Introduction and Discussion section. The revisions can be found in the Introduction section - page 2, paragraph 2, line 39-58 and Discussion section - page 12, paragraph 4, line 401-404 in the revised manuscript. |
|
Comments 2: In Figure 3, while the authors demonstrate gene integrity over five passages using PCR and viral titers, they do not report whether mCherry expression is maintained or affected by viral adaptive mutations. |
|
Response 2: Thank you for pointing this out. We agree with this comment. Therefore, the experiments to determine whether mCherry fluorescence signal changes after multiple passages are currently underway. However, the results are not yet available and therefore could not be included in this revision. In the present study, the integrity of mCherry gene insertion during serial passage was confirmed by PCR and agarose gel electrophoresis. Therefore, we inferred that the fluorescence signal is unlikely changed. |
|
Comments 3: In Figures 4 and 6, fluorescent proteins are known to be influenced by the host background, as partially indicated in Figure 4. The authors should consider examining mCherry expression in ex vivo lung tissue. |
|
Response 3: Thank you for pointing this out. We agree with this comment. In our submitted manuscript, ex vivo lung tissue was utilized to determine the correlation between mCherry fluorescence intensity and pulmonary viral load. Results demonstrated a significant correlation between mCherry fluorescence intensity and pulmonary viral load (R² = 0.69, P < 0.0001; Figure S2B). Fluorescence intensity in ex vivo lung tissue begins to increase on 2 days post-infection (dpi), peaks at 7 dpi, and subsequently declines (Figure S2A). This pattern closely parallels the in vivo fluorescence trend shown in Figure 5A. Moreover, in vivo imaging model we established was designed to enable continuous and dynamic monitoring of infection in the same animals. Examining ex vivo lung tissue at each time point would defeat the purpose of dynamic monitoring. |
|
Comments 4: Figure 4 lacks a comparison between H1N1-PR8-NSCE1-mCherry and H1N1-PR8-NSwt-mCherry, which would provide critical context for evaluating the impact of the CE1 mutation. |
|
Response 4: Thank you for pointing this out. We agree with this comment. In this study, our primary objective was to construct a recombinant influenza virus carrying mCherry and to establish an in vivo imaging model of influenza virus for antiviral drug evaluation, rather than to comprehensively delineate CE1 mutation on effects of influenza virus expressing mCherry. Compared with H1N1-PR8-NSWT-mCherry, H1N1-PR8-NSCE1-mCherry showed a higher rescue rate (Figure 1C). Meanwhile, peak titers of H1N1-PR8-NSCE1-mCherry were approximately 10.79-fold higher than H1N1-PR8-NSWT-mCherry (Figure 2B). In parallel, H1N1-PR8-NSCE1-mCherry produced detectable fluorescence signal at 12 hpi, whereas H1N1-PR8-NSWT-mCherry showed only a faint red signal even at 36 hpi (Figure 2A). Given that H1N1-PR8-NSCE1-mCherry demonstrated significantly superior viral rescue efficiency, mCherry expression levels and replication capacity compared to H1N1-PR8-NSWT-mCherry in vitro, we selected H1N1-PR8-NSCE1-mCherry for establishing the in vivo imaging model. To maintain consistency between the data and main conclusions, we have revised the Title, Abstract, Introduction, Discussion and Conclusion sections. The revisions can be found in the Title section - page 1, line 2-4; Abstract section- page 1, line 6-7, 14-15 and 20-22; Introduction section- page 2, paragraph 3, line 65-67; Discussion section- page 10, paragraph 2, line 308-311; page 11, paragraph 1, line 321-322, page 11, paragraph 2, line 338-340; Conclusion section-page 12, paragraph 5, line 408-409 and 412; page 13, paragraph 1, line 413-415 in the revised manuscript. |
|
Minor Concerns: |
|
Comments 1: The resolution of several figures is low, particularly the nucleotide sequences in Figure 1, making detailed interpretation difficult. |
|
Response 1: Thank you for pointing this out. We agree with this comment. Therefore, we have revised all figures. The revisions can be found at the following location - page 6, line 204, Figure 1; page 7, line 226, Figure 2; page 8, line 247, Figure 3; page 9, line 267, Figure 4; page 9, line 281, Figure 5; page 10, line 297, Figure 6; page 17, line 523, Figure S1 and page 17, line 530, Figure S2 in the revised manuscript. |

Reviewer 3 Report
Comments and Suggestions for Authors
Title: CE1 Compensatory Mutation Enhances Genetic Stability and Imaging Utility of an mCherry-Expressing Influenza A Virus”
Overall assessment
This manuscript presents the construction and evaluation of a genetically stable, mCherry-expressing influenza A virus incorporating the CE1 compensatory mutation. The study is clearly written, methodologically sound, and well aligned with the journal’s scope. The approach is valuable for improving live imaging tools for influenza research and antiviral screening. Overall, the work is solid and could be accepted after minor revisions.
Specific comments:
- The results clearly demonstrate improved viral stability and replication due to CE1, but it will strengthen the paper if the authors briefly discuss how CE1 may affect RNA structure or promoter activity mechanistically. Even a short comment on predicted secondary structure or known NCR behaviour would add depth.
- Include a brief discussion on the limitations of mCherry as a reporter for in vivo imaging-for instance, its limited tissue penetration and relatively low sensitivity compared to near-infrared reporters. This would show awareness of the broader context and potential improvements for future work.
- The authors might consider commenting on whether the CE1 mutation could influence host-virus interactions or immune recognition, even if indirectly, since NCR alterations might affect replication kinetics or interferon signaling.
- In Figure 2B, Figure 3B, 3C, 4B, 4C, 5A please clarify the statistical tests used and whether biological replicates were included. Something like….“Data is presented as mean ± SD/SEM of 3 independent replicates”
- It would be helpful if the fluorescence microscopy images (Figure 2A) included scale bars and possibly a merged DIC channel to better visualize infected cells.
- For Figure 4 and Figure 6, adding a timeline schematic (infection →imaging → oseltamivir treatment) could help readers grasp the workflow more intuitively.
- The discussion is clear but somewhat descriptive. It could be improved by adding a short paragraph comparing this approach to other strategies that stabilize recombinant influenza viruses (for example, promoter balancing, codon optimization, or using alternative segments for reporter insertion).
- The authors may also wish to comment on the translational potential of this model-could it be adapted for other influenza subtypes or for screening broader classes of antiviral drugs?
Writing and formatting
- The manuscript is generally well written, but a few sentences are overly long or could be made more concise. For example: “The significant correlation between fluorescence intensity and pulmonary viral load was observed…” → “A significant correlation was observed between lung fluorescence intensity and viral load.”
- Ensure consistency in the italicization of in vivo, in vitro, and gene/protein names.
- Consider softening strong conclusions like “validated CE1 mutation as a practical strategy” to “suggests that CE1 mutation represents a promising strategy.”
Author Response
|
Comments 1: The results clearly demonstrate improved viral stability and replication due to CE1, but it will strengthen the paper if the authors briefly discuss how CE1 may affect RNA structure or promoter activity mechanistically. Even a short comment on predicted secondary structure or known NCR behavior would add depth. |
|
Response 1: Thank you for pointing this out. We agree with this comment. Therefore, we have added a brief discussion on the potential mechanism by which the CE1 mutation affects RNA structure in the Discussion section. The revision can be found at the following location - page 11, paragraph 3, line 341-347 in the revised manuscript. |
|
Comments 2: Include a brief discussion on the limitations of mCherry as a reporter for in vivo imaging-for instance, its limited tissue penetration and relatively low sensitivity compared to near-infrared reporters. This would show awareness of the broader context and potential improvements for future work. |
|
Response 2: Thank you for pointing this out. We agree with this comment. Therefore, we have added a dedicated paragraph in the Discussion section to address the limitations of mCherry. The revision can be found at the following location - page 12, paragraph 3, line 386-392 in the revised manuscript. |
|
Comments 3: The authors might consider commenting on whether the CE1 mutation could influence host-virus interactions or immune recognition, even if indirectly, since NCR alterations might affect replication kinetics or interferon signaling. |
|
Response 3: Thank you for pointing this out. We agree with this comment. Therefore, we have added relevant content in the Discussion section to address the effect of CE1 mutation. The revision can be found at the following location - page 12, paragraph 3, line 347-354 in the revised manuscript. |
|
Comments 4: In Figure 2B, Figure 3B, 3C, 4B, 4C, 5A please clarify the statistical tests used and whether biological replicates were included. Something like….“Data is presented as mean ± SD/SEM of 3 independent replicates” |
|
Response 4: Thank you for pointing this out. We agree with this comment. Therefore, we have performed statistical analyses for all relevant data and revised all relevant figure legends. The revision can be found at the following location - page 6, line 209, Figure 1C; page 7, line 226 and 230 -232, Figure 2B; page 8, line 249-251, Figure 3B and 3C; page 9, line 267 and 270-273, Figure 4C and 4D; page 10, line 283, Figure 5A; page 10, line 297 and 299-304, Figure 6B and 6D; page 17, line 523 and 525-537, Figure S1; page 18, line 540, Figure S2 in the revised manuscript. |
|
Comments 5: It would be helpful if the fluorescence microscopy images (Figure 2A) included scale bars and possibly a merged DIC channel to better visualize infected cells. |
|
Response 5: Thank you for pointing this out. We agree with this comment. Therefore, we have revised the scale bar in Figure 2A. The revision can be found at the following location- page 7, line 226 and 228-229, Figure 2A. Regarding the DIC channels, we understanded the value of a merged view for presenting the environment of infected cells. However, due to current resource limitations, we were unable to incorporate DIC channels in this study. We will conduct observations using DIC channels in future studies when experimental resources permit. |
|
Comments 6: For Figure 4 and Figure 6, adding a timeline schematic (infection →imaging → oseltamivir treatment) could help readers grasp the workflow more intuitively. |
|
Response 6: Thank you for pointing this out. We agree with this comment. Therefore, we have revised Figure 4 and 6. The revision can be found at the following location - page 8, paragraph 1, line 258; page 9, line 267-269, Figure 4A and page 10, paragraph 1, line 289 and 297-299, Figure 6A in the revised manuscript. |
|
Comments 7: The discussion is clear but somewhat descriptive. It could be improved by adding a short paragraph comparing this approach to other strategies that stabilize recombinant influenza viruses (for example, promoter balancing, codon optimization, or using alternative segments for reporter insertion). |
|
Response 7: Thank you for pointing this out. We agree with this comment. Therefore, we have added the relevant content at Discussion section. The revision can be found at the following location - page 11, paragraph 2, line 323-327 in the revised manuscript. |
|
Comments 8: The authors may also wish to comment on the translational potential of this model-could it be adapted for other influenza subtypes or for screening broader classes of antiviral drugs? |
|
Response 8: Thank you for pointing this out. We agree with this comment. Therefore, we have added the relevant content in Discussion section. The revision can be found at the following location - page 11, paragraph 3, line 361-362; page 12, paragraph 1, line 363-365 and page 12, paragraph 4, line 397-404 in the revised manuscript. |
|
Comments 9: The manuscript is generally well written, but a few sentences are overly long or could be made more concise. For example: “The significant correlation between fluorescence intensity and pulmonary viral load was observed…” → “A significant correlation was observed between lung fluorescence intensity and viral load.”. |
|
Response 9: Thank you for pointing this out. We agree with this comment. Therefore, we have revised the relevant content in Abstract section. The revision can be found at the following location - page 1, line 17-18 in the revised manuscript. |
|
Comments 10: Ensure consistency in the italicization of in vivo, in vitro, and gene/protein names. |
|
Response 10: Thank you for pointing this out. We agree with this comment. Therefore, we have revised the relevant content. The revision can be found at the following location - page 5, paragraph 4, line 187 and 191; page 5, paragraph 5, line 202; page 6, paragraph 1, line 211 and 220; page 7, paragraph 2, line 244; page 11, paragraph 1, line 317 and 319 in the revised manuscript. |
|
Comments 11: Consider softening strong conclusions like “validated CE1 mutation as a practical strategy” to “suggests that CE1 mutation represents a promising strategy.” |
|
Response 11: Thank you for pointing this out. We agree with this comment. Therefore, we have revised the relevant content. The revision can be found at the following location - page 3, paragraph 2, line 83-84 in the revised manuscript. |

Reviewer 4 Report
Comments and Suggestions for Authors
The manuscript aims to improve the rescue efficiency of an mCherry-expressing influenza A virus (IAV) by introducing a CE1 mutation and to further evaluate the potential applications of this virus in antiviral drug screening. The research topic is of considerable scientific value and could contribute to the optimization of reporter IAV systems for experimental and screening purposes. However, the study presents substantial weaknesses in both experimental design and data interpretation, which need to be carefully addressed before the manuscript can be considered for publication.
Figure 2. B presents the viral growth kinetics, showing that at 36 hours post-infection, the TCIDâ‚…â‚€ of the NSwt strain is approximately one-eighth that of the NSCE1 strain. In contrast, the corresponding fluorescence intensity data in Figure 2. A display more than a 50-fold difference between the two strains. This inconsistency raises concerns about data normalization, reproducibility, and overall validity. The authors should re-examine the experimental conditions and provide clarification or additional evidence to reconcile these results.
If the primary objective of the study is to demonstrate that the CE1 mutation enhances the properties of the mCherry-expressing influenza A virus, control data for the NSwt strain should be presented in Figure 3 and in the subsequent experiments. The absence of such control data prevents a proper comparative analysis and weakens the main conclusion of the study.
Overall, while the study addresses an interesting and potentially useful topic, the current version of the manuscript requires major revisions to improve data quality, strengthen experimental rigor, and provide sufficient supporting evidence for the proposed conclusions.
Author Response
|
Comments 1: Figure 2. B presents the viral growth kinetics, showing that at 36 hours post-infection, the TCIDâ‚…â‚€ of the NSwt strain is approximately one-eighth that of the NSCE1 strain. In contrast, the corresponding fluorescence intensity data in Figure 2. A display more than a 50-fold difference between the two strains. This inconsistency raises concerns about data normalization, reproducibility, and overall validity. The authors should re-examine the experimental conditions and provide clarification or additional evidence to reconcile these results. |
|
Response 1: Thank you for pointing this out. We agree with this comment. Therefore, we have rigorously re‑examined the raw data and normalization procedures for both detection methods and consider this phenomenon to be genuine. In addition, we found that a similar phenomenon has been reported previously [1]. Zhao et al. reported that the viral titers of CE1-carrying recombinant virus PR8-NSCE1-Gluc were only slightly higher than those of recombinant virus PR8-NS-Gluc at 36 hpi (Figure 2C). However, the Gluc reporter activity in PR8-NSCE1-Gluc was more than 10-fold higher than those of PR8-NS-Gluc (Figure 3A). Gluc expression levels were determined by normalizing the bioluminescence signal to viral titer, and results showed that Gluc expression in PR8-NSCE1-Gluc-infected cells was enhanced by more than 10-fold compared with PR8-NS-Gluc-infected cells (Figure 3B). We speculated that the discrepancy may stem from the following factors. First, CE1 mutation enhances the replication and transcription of the NS segment carrying mCherry, whereas replication and transcription of the other seven segments remain unchanged. This resulted in a marked increase in mCherry expression, while the number of infectious particles (reflected by TCID50) rises only modestly. Second, mCherry is relatively stable and continuously accumulates in infected cells over time, leading to a progressive increase in fluorescence intensity. |
|
Comments 2: If the primary objective of the study is to demonstrate that the CE1 mutation enhances the properties of the mCherry-expressing influenza A virus, control data for the NSwt strain should be presented in Figure 3 and in the subsequent experiments. The absence of such control data prevents a proper comparative analysis and weakens the main conclusion of the study. |
|
Response 2: Thank you for pointing this out. We agree with this comment. In this study, our primary objective was to construct a recombinant influenza virus carrying mCherry and to establish an in vivo imaging model of influenza virus for antiviral drug evaluation, rather than to comprehensively delineate CE1 mutation on effects of influenza virus expressing mCherry. Compared with H1N1-PR8-NSWT-mCherry, H1N1-PR8-NSCE1-mCherry showed a higher rescue rate (Figure 1C). Meanwhile, peak titers of H1N1-PR8-NSCE1-mCherry were approximately 10.79-fold higher than H1N1-PR8-NSWT-mCherry (Figure 2B). In parallel, H1N1-PR8-NSCE1-mCherry produced detectable fluorescence signal at 12 hpi, whereas H1N1-PR8-NSWT-mCherry showed only a faint red signal even at 36 hpi (Figure 2A). Given that H1N1-PR8-NSCE1-mCherry demonstrated significantly superior viral rescue efficiency, mCherry expression levels and replication capacity compared to H1N1-PR8-NSWT-mCherry in vitro, we selected H1N1-PR8-NSCE1-mCherry for establishing the in vivo imaging model. To maintain consistency between the data and main conclusions, we have revised the Title, Abstract, Introduction, Discussion and Conclusion sections. The revisions can be found in the Title section - page 1, line 2-4; Abstract section- page 1, line 6-7, 14-15 and 20-22; Introduction section- page 2, paragraph 3, line 65-67; Discussion section- page 10, paragraph 2, line 308-311; page 11, paragraph 1, line 321-322, page 11, paragraph 2, line 338-340; Conclusion section-page 12, paragraph 5, line 408-409 and 412; page 13, paragraph 1, line 413-415 in the revised manuscript. |
|
Reference 1. Perez, D. R.; Zhao, X.; Lin, X.; Li, P.; Chen, Z.; Zhang, C.; Manicassamy, B.; Rong, L.; Cui, Q.; Du, R. Expanding the tolerance of segmented Influenza A Virus genome using a balance compensation strategy. PLoS Pathog. 2022, 18.
|

Round 2
Reviewer 2 Report
Comments and Suggestions for Authors
I have no further concerns and accept the manuscript in its present form